# XOR-Based Meaningful (*n*, *n*) Visual Multi-Secrets Sharing Schemes

**Sheng-Yao Huang, An-hui Lo and Justie Su-Tzu Juan ***

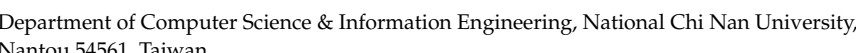

Department of Computer Science & Information Engineering, National Chi Nan University, Nantou 54561, Taiwan
* Correspondence: jsjuan@ncnu.edu.tw

**Abstract:** The basic visual cryptography (VC) model was proposed by Naor and Shamir in 1994. The secret image is encrypted into pieces, called shares, which can be viewed by collecting and directly stacking these shares. Many related studies were subsequently proposed. The most recent advancement in visual cryptography, XOR-based VC, can address the issue of OR-based VC's poor image quality of the restored image by lowering hardware costs. Simultaneous sharing of multiple secret images can reduce computational costs, while designing shared images into meaningful unrelated images helps avoid attacks and is easier to manage. Both have been topics of interest to many researchers in recent years. This study suggests ways for XOR-based VCS that simultaneously encrypts several secret images and makes each share separately meaningful. Theoretical analysis and experimental results show that our methods are secure and effective. Compared with previous schemes, our scheme has more capabilities.

**Keywords:** XOR-based visual cryptography; meaningful shares; multi-secret; secret sharing scheme; no pixel expansion

## 1. Introduction

Visual cryptographic scheme (VCS) is a secret image-sharing method. In 1994, Naor and Shamir proposed a common type of (*t*, *n*)-threshold VCS [1]. In this threshold (*t*, *n*)-VCS, a secret binary image is encrypted to get *n* images (referred to as shares), in such a way that it is impossible to learn anything about the original secret image, and then distributed sequentially to *n* participants. The secret image recovery method is straightforward; after collecting any *t* (or more) shares and stacking them together, and performing a simulated stacking operation by Boolean 'OR' operation, secret information can be revealed directly by the human visual system to identify the secret information without relying on complex calculations or cryptography knowledge. If any (*t* − 1) or fewer shares are available, then no secret will be revealed.

Early encryption methods for VCS generally used pixel expansion techniques, meaning each pixel on a secret image is expanded into *m* pixels (*m* ≥ 2) on the shares. Thus, the share size will be *m* times the size of the secret image. However, this will cause the restored image to be deformed and expanded. Kafri and Keren proposed a random grid-based visual cryptographic scheme (RGVCS) that can encrypt black and white (binary) images, which can solve the problem of pixel expansion [2]. In other words, an RGVCS creates the shares and restored image without pixel expansion. Inspired by Kafri and Keren, Shyu proposed an RGVCS for grayscale/color images [3].

Contrast is the criterion for evaluating the visual quality of the recovered secret image. Its value is between −1 and 1, with higher values indicating higher visual quality of the restored image. When the contrast value is 0, the recovered image is a meaningless image; and if the contrast value is negative, the image is a black-and-white reversed image of the original secret image. The traditional stacking operation in VCS is an OR operation, and

the contrast of the restored image in an OR-based RGVCS can only reach 0.5 at most. Such low reduction quality limits many development possibilities. To resolve the problem of poor contrast in OR-based RGVCS, XOR-based RGVCS is proposed. The visual quality of the recovered image by the XOR operation can be greatly improved because the contrast value can theoretically be equal to 1.

In 2003, Tuyls et al. introduced a new visual cryptosystem using light polarization, the operation of which is mathematically described by a XOR operation (a modulo-two addition) [4]. So that the XOR-based VCS can be implemented by using a small, cheap and lightweight decryption display, and it is more suitable for practical situations. In 2007, Wang et al. proposed a XOR-based ($n$, $n$) RGVCS [5]. Although the generated share in Wang et al.'s paper is not pixel-expanded and perfectly displays the secret image, some issues are still refined. One of them is that the shares are meaningless, which will attract the attention of cybercriminals and make it hard to manage.

In 2013, Wu and Sun proposed a (2, 2) generalized XOR-based RGVCS [6], where the average light transmission of a share becomes adjustable. In their study, the visual quality of the shares and recovered image are still not good enough due to the limitations of the design method. In order to facilitate management and avoid unnecessary suspicion and attacks, VCS has added a new concept to make shares meaningful in recent years. Its main purpose is to make shares that look like a random grid no longer meaningless, and users can easily identify who is who with the naked eye. In 2015, Ou et al. proposed a ($n$, $n$) XOR-based RGVCS with meaningful shares [7] for increasing the image quality. They define a variable $\beta$ to balance the visual quality of the restored image with shares. As the value of $\beta$ becomes larger, the visual quality of the restored image will increase, while the camouflage result of the share will decrease. Furthermore, their method provides perfect black pixel reconstruction, which makes the restored image more recognizable by human vision. In 2021, Lo and Juan proposed three ($n$, $n$) XOR-based RGVCS [8] to improve Ou et al.'s method. The shares created in the scheme of Ou et al. must have the same camouflage. Using Lo and Juan's scheme allows shares to have various camouflage objects. They are both methods of encrypting one secret image at a time.

In comparison to the VCS mentioned above, the visual multi-secret sharing scheme (VMSSS) has the ability to simultaneously encrypt numerous secrets into shares. As a result, it can reduce certain extra expenses while improving the encryption's performance. In recent years, various study projects on VMSSS have been made, including ([9–13]). In 2008, Chen et al. proposed a four-secrets sharing scheme [11]. This method is encrypting four secret images into the shares, and one can rotate one share by 0, 90, 180, and 270 degrees and stack it on the other share to restore the four secret images, respectively. However, this approach can only encrypt four square secrets at the same time. To break through the constraints on the number and shape of secret images, Chang et al. in 2018 proposed a new VMSSS via random grids [9]. One secret image is first divided into numerous fragments, each of which is then independently encrypted to the associated share. By moving one share 0, $w/p$, $2w/p$, $3w/p$, ..., $(N-1)w/p$ pixels and stacking it on another share, the first, second, third, ..., $N$th secret images are recovered, where $w$ is the width of the image.

In this paper, we propose four ($n$, $n$) XOR-based VMSSSs (or XOR-based RGVCSs) that can simultaneously encrypt more than one secret image and separately make each share disguised as a meaningful image. The rest of this paper is organized as follows. Section 2 shows the related work. Section 3 gives the proposed scheme and some experimental results. Some analyses are presented in Section 4. The conclusion and future work are given in Section 5.

## 2. Related Work

In this paper, we set a pixel to be 1 when it is black and 0 when it is white. To help understand the proposed scheme, some related VCSs are introduced in this section.

### 2.1. Random Grid-Based Visual Cryptography Scheme

Kafri and Keren proposed three basic RGVCS in 1987 [2]. In a random grid (RG), each pixel can be either completely transparent (white) or completely opaque (black), and the choice between the two options is chosen by a coin toss. There is no correlation between the values of the different pixels in the array. They used $S(i, j)$ to represent a pixel in the image $S$ and defined that $S(i, j) = 1$ when a pixel is black (opaque) and $S(i, j) = 0$ when a pixel is white (transparent). They use Boolean ORs to compute "stacked" operations because the results are close to human vision. The three basic RGVCS are as follows.

---

**Algorithm KK1.** [2]

---

**Input:** The secret image $S$ with size $w \times h$ pixels.
**Output:** Two shares $G_1$ and $G_2$ with size $w \times h$.

1.  Generate a $w \times h$ random grid $G_1$.

2.  **For** $i = 0$ **to** $w - 1$ **do**
    **For** $j = 0$ **to** $h - 1$ **do**
    **If** $(S(i, j) == 0)$
    **then** $G_2(i, j) = G_1(i, j)$;
    **else** $G_2(i, j) = \overline{G_1(i, j)}$;

3.  **Return** $G_1$ and $G_2$.

---

**Algorithm KK2.** [2]

---

**Input:** The secret image $S$ with size $w \times h$ pixels.
**Output:** Two shares $G_1$ and $G_2$ with size $w \times h$.

1.  Generate a $w \times h$ random grid $G_1$.

2.  **For** $i = 0$ **to** $w - 1$ **do**
    **For** $j = 0$ **to** $h - 1$ **do**
    **If** $(S(i, j) == 0)$
    **then** $G_2(i, j) = G_1(i, j)$;
    **else** $G_2(i, j) = \text{random}(0, 1)$;

3.  **Return** $G_1$ and $G_2$.

---

**Algorithm KK3.** [2]

---

**Input:** The secret image $S$ with size $w \times h$ pixels.
**Output:** Two shares $G_1$ and $G_2$ with size $w \times h$.

1.  Generate a $w \times h$ random grid $G_1$.

2.  **For** $i = 0$ **to** $w - 1$ **do**
    **For** $j = 0$ **to** $h - 1$ **do**
    **If** $(S(i, j) == 0)$
    **then** $G_2(i, j) = \text{random}(0, 1)$;
    **else** $G_2(i, j) = \overline{G_1(i, j)}$;

3.  **Return** $G_1$ and $G_2$.

---

### 2.2. XOR-Based Visual Secret Sharing Scheme with Meaningful Shares

An $(n, n)$-threshold XOR-based VCS with meaningful shares was proposed by Ou et al. in 2015 [7]. Their encryption scheme includes three algorithms that may encrypt a secret image $S$ into $n$ meaningful shares. We briefly introduce these three algorithms as follows.

In their Algorithm 1, a matrix $M_n$ is formed. $M_n$ is a $2^n \times n$ matrix, and the element in row $i$ is the binary representation of $i - 1$. Then, partition $M_n$ into two sub-matrices, $M_n^{odd}$ and $M_n^{even}$, such that for each row vector in $M_n^{odd}$ ($M_n^{even}$, respectively), the hamming weight is odd (even, respectively). Their Algorithm 2 gave the basic algorithm for a $(n, n)$

XOR-based VCS: For each position $(i, j)$ in the secret image $S$, if $S(i, j) = 0$ (1, respectively), construct $n$ share pixels $R_1(i, j), \ldots, R_n(i, j)$. by randomly choosing a row vector $r$ of the matrix $M_n^{even}$. ($M_n^{odd}$, respectively) then assign the value of the $(r, k)$ element in the matrix to $R_k(i, j)$. In their Algorithm 3, for each position $(i, j)$ in the secret image $S$, they generate a random bit $d$, which is 1 with probability $\beta$ at first. If $d = 1$, the shares are equal to $R_1(i, j)$, $\ldots, R_n(i, j)$. Otherwise, let the pixel of each share be the value of the cover image $C(i, j)$, except for a random share if $n \times C(i, j) = 0 \mod 2$.

We will design our VCS using the idea of their VCS, encrypting the secret image into $n$ meaningful shares.

### 2.3. Visual Multiple Secrets Sharing Scheme by Random Grids

A (2, 2)-visual multiple secret sharing scheme (VMSSS) was proposed by Chang et al. [9] in 2018. They started by defining three functions $f_p$, $f_{RG}$ and $f_{ORG}$ in order to simultaneously encrypt $N$ secret images. Function $f_p$ randomly chose a pixel from $S$; Function $f_{RG}$ is given for the first encryption; and Function $f_{ORG}$ is given for the rest of the encryption. The main idea of Chang et al.'s VMSSS is to evenly encrypt any two consecutive secret images. In their algorithm, the function $f_p$ is first used to randomly select a pixel from the pair of images. Then, randomly select a pair of consecutive secret images ((0, 1), (1, 2), $\ldots$, or (N–1, 0)). Then, the selected pixel pair is encrypted by the functions $f_{RG}$ and $f_{ORG}$. Repeat the above steps until all pixels in all shares are generated.

The paper defines distortion as the unencrypted ratio in the algorithm. The distortion of their algorithm is $((N-2)p + 1)/Np$, where $p$ is selected by the user, which must be the divisor of $w$. Besides, $N - 1$ and $p$ must be mutually prime.

We will refer to their scheme of encrypting multiple secret images into 2 shares at the same time as our first step.

## 3. Main Scheme

In this section, based on a random grid, we want to simultaneously encrypt multiple secret images into $n$ meaningful shares, and only collect all $n$ shares and use XOR as the operation to recover all secret images. We will give four (n, n) threshold VMSSSs with meaningful shares to separately solve the same problem in the following sections. Their models are all similar; they only differ in a key step—Algorithm III. So, we will firstly introduce the models of the encryption and decryption.

### 3.1. The Process and Definition of the Proposed Schemes

Table 1 provides illustrations of some of the symbols and parameters used in this paper to aid in understanding the proposed VCS.

**Table 1.** Some symbols and parameters used in this paper.

| Notation | Description |
| --- | --- |
| 0 | A white pixel |
| 1 | A black pixel |
| $S_i$ | The $i$-th secret image |
| $D_i$ | The $i$-th camouflage image |
| $C_i$ | The $i$-th meaningless image |
| $M_i$ | The $i$-th (meaningful) share |
| $R_i$ | The $i$-th reconstructed image |
| $S(i, j)$ | The value of the pixel at position $(i, j)$ in image $S$ |
| $N$ | The number of secret images |
| $n$ | The number of shares and camouflaged images |
| $p$ | The number of pieces that an image will be divided into |

The process of our schemes all consist of three encryption algorithms and one decryption algorithm. The models for the proposed encryption and decryption procedures are shown in Figures 1 and 2, respectively.

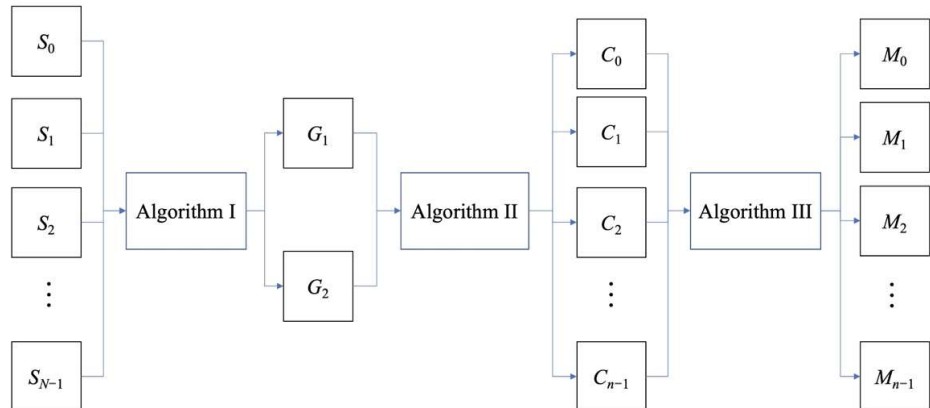

**Figure 1.** The model of the encryption process of the proposed scheme.

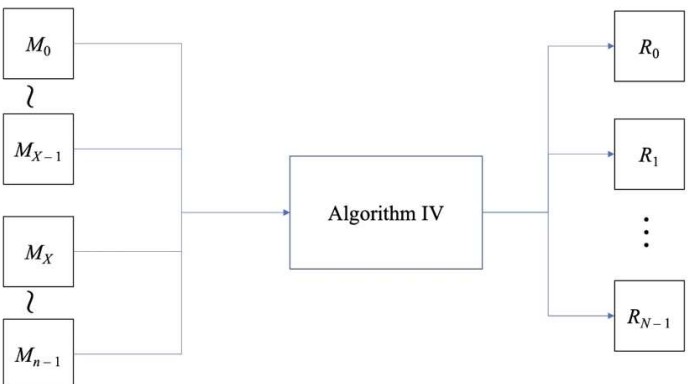

**Figure 2.** The model of the decryption process of the proposed scheme.

### 3.2. Algorithm I. XOR-Based Visual Multi-Secret Scheme

We apply a XOR operation rather than OR when encrypting input images in our Algorithm I since Chang et al.'s [9] served as inspiration. We were able to successfully enhance Chang et al.'s scheme's restoring effect as a consequence. Here, we modify the two original functions—$f_{RG}$ and $f_{ORG}$—to create $f_{XRG}$ and $f_{XORG}$. Actually, $f_{XRG}$ can be selected from any one of the three random grid algorithms in [2] (as we described in Section 2.1), which inputs a pixel of the secret image, then outputs two cipher-pixels for two shares. $f_{XORG}$ is the function based on $f_{XRG}$, which inputs a cipher-pixel of shares and a pixel of the secret image, then outputs the other cipher-pixel. Although these three random grid algorithms (KK1, KK2, and KK3 in Section 2.1) in [2] are computed in the OR-operation, it is not difficult to see that they also apply to the XOR-operation. Therefore, there are also three possible design methods for the function $f_{XRG}$ and $f_{XORG}$. We use the one with the best contrast in the recovered image, listed below.

---

**Function $f_{XRG}$**

---

**Input:** The pixel of the secret image $S(i, j)$.
**Output:** The pixels of the two shares $G_1(i, j)$ and $G_2(i, j)$.

1.     $G_1(i, j) = \text{random}(0, 1)$;

2.     **If** $(S(i, j) == 0)$
            **then** $G_2(i, j) = G_1(i, j)$;
      **else** $G_2(i, j) = \overline{G_1(i, j)}$;

3.     **Return** $G_1(i, j)$ and $G_2(i, j)$.

---

---

**Function** $f_{XORG}$

---

**Input:** Two pixels of the secret image $S(i, j)$ and one share $G_1(i, j)$.
**Output:** A pixel of the other share $G_2(i, j)$.

1.    **If** $(S(i, j) == 0)$
          **then** $G_2(i, j) = G_1(i, j)$;
      **else** $G_2(i, j) = \overline{G_1(i, j)}$;

2.    **Return** $G_2(i, j)$.

---

Algorithm 1, shown below (for Algorithm I), has the goal of simultaneously encrypting multiple secret images into two shares. The purpose is to shift one of the shares by $i$ unit and superimpose it back into another share to restore $i$th secret image. Before starting Algorithm 1, we need to number the $N$ secret images $S_0, S_1, S_2, \ldots, S_{N-2}, S_{N-1}$. Then, we assign two consecutive secret images as pairs $(S_0, S_1), (S_1, S_2), \ldots, (S_{N-2}, S_{N-1}), (S_{N-1}, S_0)$. In a secret image of size $w \times h$, the secret image is split into $p$ parts, which means that each shift is in units of $(w/p)$ pixels. In addition, the value of $p$ must be mutually prime with $N-1$; otherwise, when generating the share, the value of the same position will be repeatedly generated in the loop, causing the stacked share to be unsuccessful in recovering the secret image. We randomly chose any two consecutive secret images $(S_A, S_{A+1})$ to encrypt each pixel. Since when $A = N-1$, the last secret image, the next must be the first secret image, 0. The formula for the variables will be different in this case than in the other cases, so Step 5 must be separately presented. The schematic diagram of Algorithm 1 is shown in Figure 3.

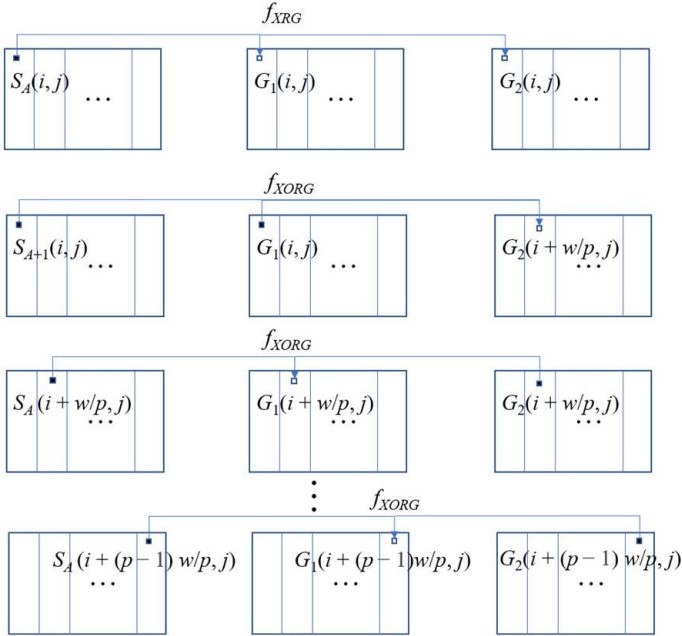

**Figure 3.** The schematic diagram of Algorithm 1 when $A = 0$.

In Step 1 of Algorithm 1, we first select a random secret image $S_A$ from $N$ secret images, and Step 2 select a fresh pixel $G_1(i, j)$. Step 3 generates two pixels by the Function $f_{XRG}$. The first pixel $G_1(i, j)$ is at the same position as $S_A(i, j)$. The second pixel $G_2(i + A \times (w/p), j)$ is shifted by $A \times (w/p)$ pixels for $S_A(i, j)$. The next encryption process will be divided into two parts, depending on whether the selected secret image $S_A$ is the last secret image or not. If $S_A$ is the last secret image ($S_A = S_{N-1}$), the secret $S_{A+1}$ encrypted with $S_A$ will be out of range, so we set it to be $S_0$. In this case, the calculation will be different from the normal case. If the selected secret image $S_A$ is not the last one ($S_A = \{S_0, S_1, S_2, \ldots, S_{N-2}\}$), go to Step 4. The pixel $G_2((i + A' \times w/p) \bmod w, j)$ is generated by the Function $f_{XORG}$ based on another secret image $S_{A+1}$ of the same group with the share $G_1(i, j)$. Then, the pixel $G_2((i +$

$A' \times w/p$) mod $w$, $j$) is used with the known secret image pixel $S_A((i + (k + 1) \times w/p)$ mod $w$, $j$) to generate pixel $G_1((i + (k + 1) \times w/p)$ mod $w$, $j$), and the process is repeated $p$ times. If the selected secret image $S_A$ is the last one ($S_A = S_{N-1}$), go to Step 5. We generate a group of pixels on share $G_2$ based on another secret image $S_0$ and the associated pixels on share $G_1$. Further, we use the generated pixels on $G_2$ to encrypt the associated pixels on $G_1$ by $S_{N-1}$. Repeat this step $p$ times to finish. Finally, repeat the above Step 1–Step 5 until all the pixels in the shares $G_1$ and $G_2$ are generated.

---

**Algorithm 1.**

---

**Input:** N secret images $S_0$, $S_1$, ..., $S_{N-1}$ with size $w \times h$ pixels, a positive integer $p$ (must be a divisor of $w$, and coprime to $N - 1$).
**Output:** Two shares $G_1$ and $G_2$ with size $w \times h$.

1.  $A$ = random (0, $N - 1$)

2.  $S_A(i, j) \leftarrow f_p(S_A)$

3.  Generate two pixels $G_1(i, j)$ and $G_2((i + A \times w/p) \% w, j)$ by Function $f_{XRG}(S_A(i,j))$.
    **If** $A \in \{0, 1, \ldots, N - 2\}$,
        **then** go to Step 4.
    **else** $(A = N - 1)$,
        go to Step 5.

4.  **For** $k = 0$ **to** $p - 1$ **do**
    Let $A' = (A + k + 1) \% p$.
    $G_2((i + A' \times w/p) \% w, j) = f_{XORG}(S_{A+1}((i + k \times w/p) \% w, j),$
        $G_1((i + k \times w/p) \% w, j))$.
    $G_1((i + (k+1) \times w/p) \% w, j) = f_{XORG}(S_A((i + (k+1) \times w/p) \% w, j),$
        $G_2((i + A' \times w/p) \% w, j))$.

5.  **For** $k = 0$ **to** $p - 1$ **do**
    Let $A'' = 1 - N$.
    $G_2((i + A'' \times w/p) \% w, j) = f_{XORG}(S_0((i + k \times A'' \times w/p) \% w, j),$
        $G_1((i + k \times A'' \times w/p) \% w, j))$.
    $G_1((i + (k+1) \times A'' \times w/p) \% w, j) = f_{XORG}(S_A((i + (k+1) \times A'' \times w/p) \% w, j),$
        $G_2((i + k \times A'' \times w/p) \% w, j))$

6.  **Repeat** Steps 1–5 until all pixels of two shares $G_1$ and $G_2$ are processed.

---

Steps 4–5 including $O(p)$ commands, which will be executed $w \times h/p$ times. So the time complexity of Algorithm 1 is $O(w \times h)$.

### 3.3. Algorithm II. Augmented Shares Scheme

Algorithm II extends the two shares generated by Algorithm I to $n$ shares, so that $n$ participants can share the secret image together. They will keep their own share, so the secret image cannot be recovered without any of them during recovery. In fact, Algorithm II directly encrypts $G_1$ into $\lfloor n/2 \rfloor$ shares and $G_2$ into $\lfloor n/2 \rfloor$ shares. After calculating the number of $X$, $Y$, we start to pick a pixel $(i, j)$ and assign values to all of the shares $C_0 \sim C_{n-1}$ in the same pixel position. As with coin tosses, the value is set to 0 or 1, with a 1/2 chance. Next, we calculate the sum of $C_0(i, j) \sim C_{X-1}(i, j)$ as $s$, and $C_X(i, j) \sim C_{n-1}(i, j)$ as $t$. In the process of giving values to the first part of the share, we first determine whether $G_1(i, j)$ is white or not. If it is white (= 0), and the sum $s$ is odd, we also randomly select $C_x$ from these $X$ shares; let $C_x(i, j) = \overline{C_x}(i, j)$. If they add up to an even number, their values are not changed. If $G_1(i, j)$ is black (= 1), and if the sum $s$ is even, we randomly select $C_x$ from these $X$ shares; let $C_x(i, j) = \overline{C_x}(i, j)$ again. If $s$ is an odd number, their values are not changed. The second part of the share is calculated in the same way as the first part. We present Algorithm 2 for Algorithm II as shown below. The time complexity of Algorithm 2 is also O($whn$).

---

**Algorithm 2.**

---

**Input:** Two shares $G_1$ and $G_2$ with size $w \times h$, and a positive integer $n \geq 2$.

**Output:** $n$ shares $C_0, C_1, \ldots, C_{n-1}$ with size $w \times h$.

1.   Let $X = \lceil n/2 \rceil$ and $Y = \lceil n/2 \rceil$.

2.   **For** each pixel $(i, j)\{0 \leq i \leq w, 0 \leq j \leq h\}$, **do**

3.      $x = \text{random} (0, X-1), y = \text{random} (0, Y-1)$.

4.      $C_0(i, j), \ldots, C_{n-1}(i, j) = \text{random}(0, 1)$, and calculate $s = C_0(i, j) + \ldots + C_{X-1}(i, j)$, $t = C_X(i, j) + \ldots + C_{n-1}(i, j)$.

5.      **If** $(G_1(i, j) = 0$ and s is odd) or $(G_1(i, j) = 1$ and s is even) **then** $C_x(i, j) = \overline{C_x (i, j)}$.

6.      **If** $(G_2(i, j) = 0$ and t is odd) or $(G_2(i, j) = 1$ and t is even) **then** $C_{X+y}(i, j) = \overline{C_{X+y} (i, j)}$.

7.   **Return** $C_0, C_1, \ldots, C_{n-1}$.

---

### 3.4. Algorithm III. Meaningful Shares Scheme

We try to give the shares produced by Algorithm II with meaning in this subsection. To specify the ratio of a share that is required to be covered, we create the parameter $\gamma$ ($0 \leq \gamma \leq 1$) in this case. When $d = 0$ (with probability $1-\gamma$), Algorithm III directly copies the output pixels of Algorithm II as the final output share. When $d = 1$ (with probability $\gamma$), Algorithm III first copies the camouflaged pixels into the final output shares, and then encrypts them to remove the disguised parts when decrypting. When $\gamma = 1$, it means that the whole share is camouflaged. When $\gamma = 0.5$, it means that the proportion of pixels that are camouflaged in a share is about 50% of all pixels, and the proportion of pixels that are not camouflaged is 50% of all pixels. When $\gamma = 0$, it means that the whole share has no disguise. The size of the $\gamma$ also affects the quality of the restored image, and the flexible adjustment of $\gamma$ makes it easier for users to make decisions. Therefore, as $\gamma$ increases, more of the shares are camouflaged. We design four different methods for Algorithm III so that users can choose the method that best satisfies their needs depending on the condition. Algorithm 3 represents Method 1, called average encryption; the other three methods will use it as the basic idea of the extension. The flow chart for Method 1 Steps 1–4 is shown in Figure 4.

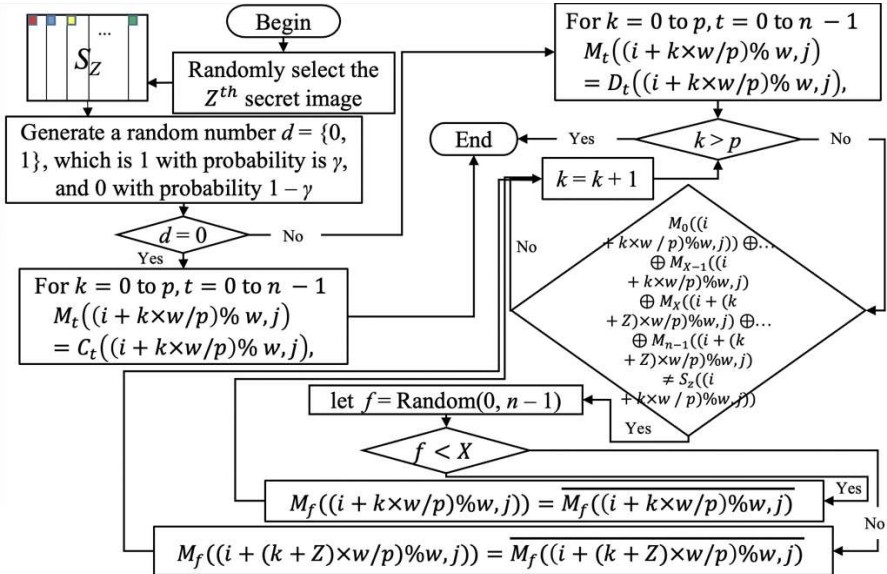

**Figure 4.** The flow chart of Method 1 Steps 1–4.

In the most important step (main different step with other methods), Step 4 when $d = 1$, we must encrypt the camouflage image $D$ into the share $M$. First, give the value of the camouflage image to the meaningful sharing image $M_{0\sim n-1}((i + k \times w/p) \bmod w, j) = D_{0\sim n-1}((i + k \times w/p) \bmod w, j)$ for $k = 0$ to $p$. Next, we must calculate whether the result of the Boolean XOR operation on the shares $M_0 \sim M_{n-1}$ is the same as the selected $Z$th secret image $S_Z((i + k \times w/p) \bmod w, j)$. If it is different, proceed with the following calculation: let $X$ is $?n/2?$, and we randomly select one $M_f$ among $n$ meaningful sharing images. If the selected image is among $M_0 \sim M_{X-1}$, let the new value of $M_f((i + k \times w/p) \bmod w, j)$ to be $\overline{M_f((i + k \times w/p) \bmod w, j)}$. If the selected $M_f$ is among $M_X \sim M_{n-1}$, let the new value of $M_f((i + (k + Z) \times w/p) \bmod w, j)$ be $\overline{M_f((i + (k + Z) \times w/p) \bmod w, j)}$. Executing $p$ times generates the pixels in these positions. $k$ has an initial value of 0 and is incremented by 1 each time it finishes.

---

**Algorithm 3. [Method 1] Average encryption**

---

**Input:** $N$ secret images $S_0, S_1, \ldots, S_{N-1}$, $n$ meaningless shares $C_0, C_1, \ldots, C_{n-1}$, $n$ camouflage images $D_0, D_1, \ldots, D_{n-1}$, all of them with size $w \times h$, and a positive integer $p$.

**Output:** $n$ meaningful shares $M_0, \ldots, M_{n-1}$ with size $w \times h$.

1. Let $X = \lceil n/2 \rceil$, $Y = \lceil n/2 \rceil$, and randomly select a pixel $(i, j)\{0 \le i \le w, 0 \le j \le h\}$.

2. $Z = \text{random}(0, N-1)$.

3. Define a number $d = \text{random}(0, 1)$, which is 1 with probability $\gamma$.

4. **If** $d = 0$ **then**
      **For** $k = 0$ **to** $p$ **do**
        **For** $r = 0$ **to** $n - 1$ **do**.
         $M_r((i + k \times w/p) \% w, j) = C_r((i + k \times w/p) \% w, j)$
      **else**
      **For** $k = 0$ **to** $p$ **do**
        **For** $r = 0$ **to** $n - 1$ **do**
         $M_r((i + k \times w/p) \% w, j) = D_r((i + k \times w/p) \% w, j)$.
      **For** $k = 0$ **to** $p$ **do**
        **If** $(M_0((i + k \times w/p)\%w, j) \oplus \ldots \oplus M_{X-1}((i + k \times w/p)\%w, j) \oplus M_X((i + (k + Z) \times w/p)\%w, j) \oplus \ldots \oplus M_{n-1}((i + (k + Z) \times w/p)\%w, j) \ne S_z((i + k \times w/p)\%w, j))$, **then**
          Let $f = \text{random}(0, n - 1)$.
          **If** $(f < X)$ **then**
           $M_f((i + k \times w/p) \% w, j) = \overline{M_f((i + k \times w/p) \% w, j)}$.
          **else**
           $M_f((i + (k + Z) \times w/p) \% w, j)) = \overline{M_f((i + (k + Z) \times w/p) \% w, j)}$.

5. **Repeat** Steps 1–4 until all pixels are processed in $n$ meaningful shares, then **return** $M_0, M_1, \ldots, M_{n-1}$.

---

The time complexity of Method 1 is $O(whn)$. Next, we will list some experimental results to better understand each of our proposed methods. Figure 5 shows two secret images, and five camouflage images used in the following experiments, and Figure 6 shows the experimental results of Method 1. The size of each image is $720 \times 720$ pixels.

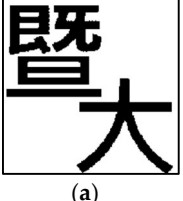 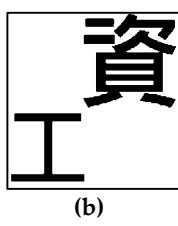 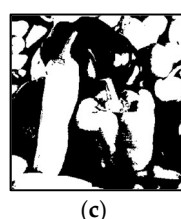 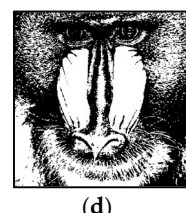 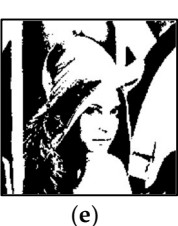 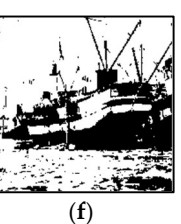 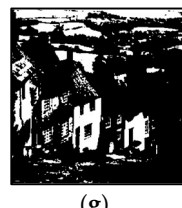

(**a**)　　(**b**)　　(**c**)　　(**d**)　　(**e**)　　(**f**)　　(**g**)

**Figure 5.** (**a**,**b**) Secret images $S_0$, $S_1$, (**c–g**) Camouflage images $D_0, \ldots, D_4$.

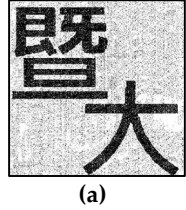 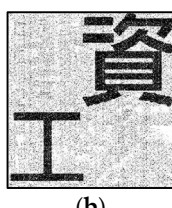 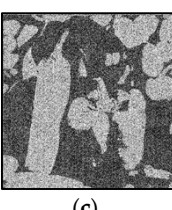 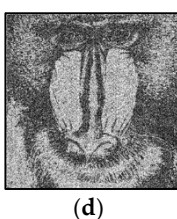 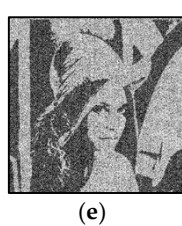 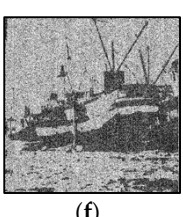 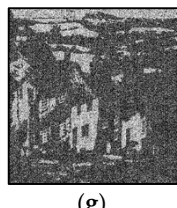

(**a**)　　(**b**)　　(**c**)　　(**d**)　　(**e**)　　(**f**)　　(**g**)

**Figure 6.** Experimental results by Method 1 for (5, 5) with $p = 45$, $\gamma = 0.5$. (**a**,**b**) Restored images $R_0$, $R_1$, (**c–g**) Shares $M_0, \ldots, M_4$.

In Method 1, the white part of the secret images in the restored images are affected by the camouflaged images. Therefore, in order to remove the influence of the camouflage images, we make the pixels involved in camouflage appear black when restored in Method 2: Enhanced encryption. Algorithm 4 shows Method 2 as follows.

---

**Algorithm 4. [Method 2] Enhanced encryption**

---

**Input:** $N$ secret images $S_0$, $S_1$, …, $S_{N-1}$, $n$ meaningless shares $C_0$, $C_1$, …, $C_{n-1}$, $n$ camouflage images $D_0$, $D_1$, …, $D_{n-1}$, all of them with size $w \times h$, and a positive integer $p$.

**Output:** $n$ meaningful shares $M_0, \ldots, M_{n-1}$ with size $w \times h$.

1.　Let $X = \lceil n/2 \rceil$, $Y = \lceil n/2 \rceil$, and randomly select a pixel $(i, j)\{0 \le i \le w,\ 0 \le j \le h\}$.

2.　$Z = $ random $(0,\ N - 1)$.

3.　Define a number $d = $ random $(0,\ 1)$, which is 1 with probability $\gamma$.

4.　**If** $d = 0$ **then**
　　**For** $k = 0$ **to** $p$ **do**
　　　**For** $r = 0$ **to** $n - 1$ **do**
　　　　$M_r((i + k \times w/p) \% w, j) = C_r((i + k \times w/p) \% w, j)$.
　　**else**
　　　**For** $k = 0$ **to** $p$ **do**
　　　　**For** $r = 0$ **to** $n - 1$ **do**
　　　　　$M_r((i + k \times w/p) \% w, j) = D_r((i + k \times w/p) \% w, j)$.
　　　**For** $k = 0$ **to** $p$ **do**
　　　　**If** $(M_0((i + k \times w/p)\%w, j) \oplus \ldots \oplus M_{X-1}((i + k \times w/p)\%w, j) \oplus M_X((i + (k + Z) \times w/p)\%w, j) \oplus \ldots \oplus M_{n-1}((i + (k + Z) \times w/p)\%w, j) \ne 1)$, **then**
　　　　　Let $f = $ random$(0, n - 1)$.
　　　　　**If** $(f < X)$ **then**
　　　　　　$M_f((i + k \times w/p) \% w, j) = \overline{M_f((i + k \times w/p) \% w, j)}$.
　　　　　**else**
　　　　　　$M_f((i + (k + Z) \times w/p) \% w, j)) = \overline{M_f((i + (k + Z) \times w/p) \% w, j)}$.

5.　**Repeat** Steps 1–4 until all pixels are processed in $n$ meaningful shares, then **return** $M_0, M_1, \ldots, M_{n-1}$.

---

The difference between Method 1 and Method 2 is the change in statement in Step 4, so the time complexity of Method 2 is still $O(whn)$. Figure 7 shows the flow chart of Method 2 Steps 1–4, and Figure 8 shows the experimental results of Method 2.

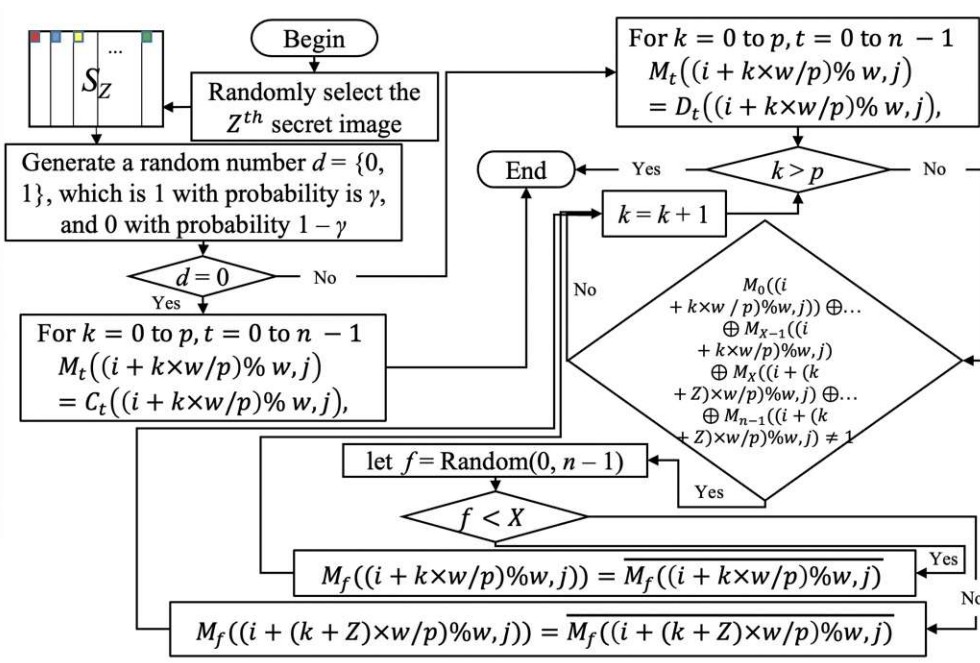

**Figure 7.** The flow chart of Method 2 Steps 1–4.

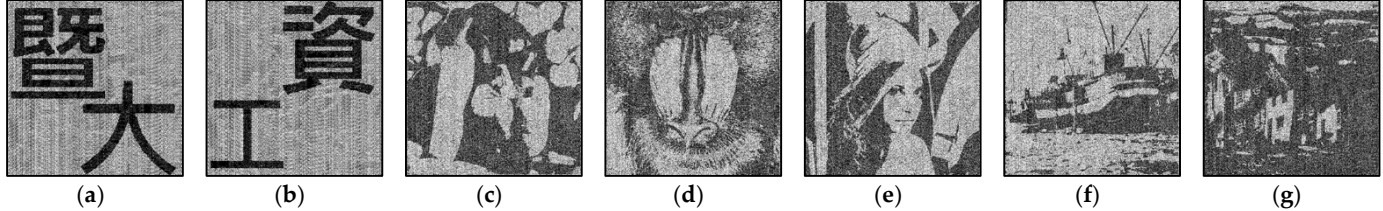

**Figure 8.** Experimental results by Method 2 for (5, 5) with $p = 45$, $\gamma = 0.5$. (**a**,**b**) Restored images $R_0$, $R_1$ (**c**–**g**) Shares $M_0, \ldots, M_4$.

Next, we introduce Method 3: Favor encryption, which focuses on encrypting the first secret image so that the first secret image is as completely restored as possible. In Methods 1 and 2, although each recovered image is partially recoverable, the chance of each secret being chosen to be recoverable decreases as the number of $N$ increases. Sometimes the user needs to specify that a certain secret image has a higher restored visual quality. We propose Method 3 to solve this problem, and the restoration of this image is as good as in Algorithm I. However, other unselected secret images will be slightly less recovered. Algorithm 5 shows Method 3 as follows.

The difference between Method 3 and the previous two methods is that we delete the previous Step 2 and modify Step 3. The time complexity of Method 3 is also $O(whn)$. The flow chart of Method 3 Steps 1–3 is shown in Figure 9, and the experimental results of Method 3 is illustrated in Figure 10.

---

**Algorithm 5. [Method 3] Favor encryption**

---

**Input:** $N$ secret images $S_0, S_1, \ldots, S_{N-1}$, $n$ meaningless shares $C_0, C_1, \ldots, C_{n-1}$, $n$ camouflage images $D_0, D_1, \ldots, D_{n-1}$, all of them with size $w \times h$, and a positive integer $p$.

**Output:** $n$ meaningful shares $M_0, \ldots, M_{n-1}$ with size $w \times h$.

1. Let $X = \lceil n/2 \rceil$, $Y = \lceil n/2 \rceil$, and randomly select a pixel $(i, j)\{0 \le i \le w, \ 0 \le j \le h\}$.

2. Define a number $d = $ random $(0, 1)$, which is 1 with probability $\gamma$.

3. **If** $d = 0$ **then**
   **For** $k = 0$ **to** $p$ **do**
   　**For** $r = 0$ **to** $n - 1$ **do**
   　　$M_r((i + k \times w/p) \ \% \ w, j) = C_r((i + k \times w/p) \ \% \ w, j)$.
   **else**
   **For** $k = 0$ **to** $p$ **do**
   　**For** $r = 0$ **to** $n - 1$ **do**
   　　$M_r((i + k \times w/p) \ \% \ w, j) = D_r((i + k \times w/p) \ \% \ w, j)$.
   **For** $k = 0$ **to** $p$ **do**
   **If** $(M_0((i + k \times w/p)\%w, \ j) \oplus \ldots \oplus M_{n-1}((i + k \times w/p)\%w, \ j) \ne$
   $S_0((i + k \times w/p)\%w, \ j))$, **then**
   　Let $f = $ random $(0, n - 1)$.
   　$M_f((i + k \times w/p) \ \% \ w, \ j) = \overline{M_f((i + k \times w/p) \ \% \ w, \ j)}$.

4. **Repeat** Steps 1–3 until all pixels are processed in $n$ meaningful shares, then
   **return** $M_0, M_1, \ldots, M_{n-1}$.

---

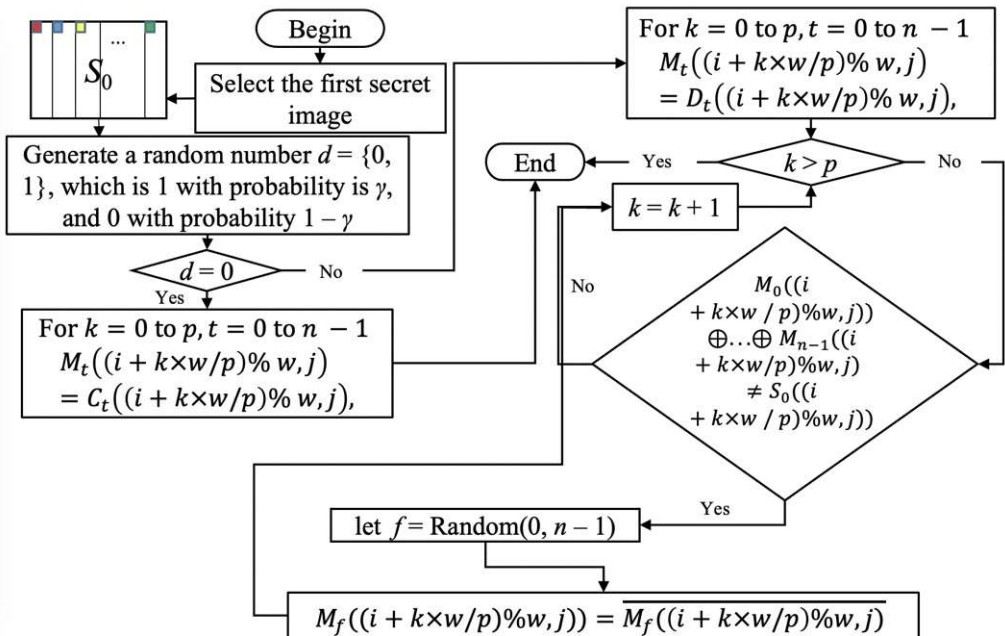

**Figure 9.** The flow chart of Method 3 Steps 1–3.

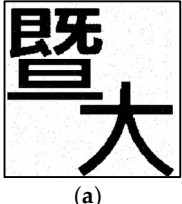 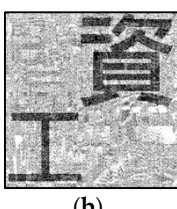 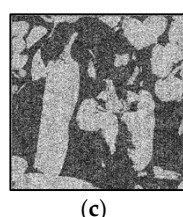 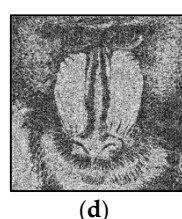 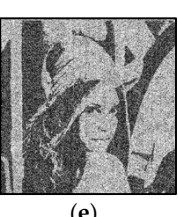 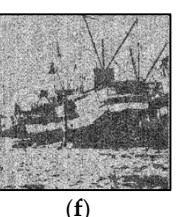 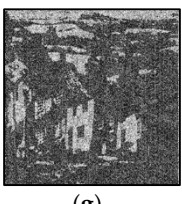

(a)      (b)      (c)      (d)      (e)      (f)      (g)

**Figure 10.** Experimental results by Method 3 for (5, 5) with $p = 45$, $\gamma = 0.5$. (**a**,**b**) Restored images $R_0$, $R_1$ (**c–g**) Shares $M_0, \ldots, M_4$.

Last, we would like to introduce Method 4. Because Method 2 only considers whether the result is black pixels for the $Z^{th}$ image, it is unable to fully restore all the black pixels for all secret images. In order to restore the black pixels of all secret images completely (perfect black), we propose Method 4: Perfect black encryption. Algorithm 6 shows Method 4 as follows.

---

**Algorithm 6. [Method 4] Perfect black encryption**

---

**Input:** $N$ secret images $S_0$, $S_1$, $\ldots$, $S_{N-1}$, $n$ meaningless shares $C_0$, $C_1$, $\ldots$, $C_{n-1}$, $n$ camouflage images $D_0$, $D_1$, $\ldots$, $D_{n-1}$, all of them with size $w \times h$, and a positive integer $p$.
**Output:** $n$ meaningful shares $M_0, \ldots, M_{n-1}$ with size $w \times h$.

1.  Let $X = \lceil n/2 \rceil$, $Y = \lceil n/2 \rceil$, and randomly select a pixel $(i, j)\{0 \leq i \leq w, 0 \leq j \leq h\}$.

2.  Define a number $d$ = random $(0, 1)$, which is 1 with probability $\gamma$.

3.  **If** $d = 0$ **then**
    **For** $k = 0$ **to** $p$ **do**
      **For** $r = 0$ **to** $n - 1$ **do**
        $M_r((i + k \times w/p) \ \% \ w, j) = C_r((i + k \times w/p) \ \% \ w, j)$.
    **else**
      **For** $k = 0$ **to** $p$ **do**
        **For** $r = 0$ **to** $n - 1$ **do**
          $M_r((i + k \times w/p) \ \% \ w, j) = D_r((i + k \times w/p) \ \% \ w, j)$.
    **For** $k = 0$ **to** $p$ **do**
      **If** $M_0((i + k \times w/p)\%w, \ j)) \oplus \ldots \oplus M_{X-1}((i + k \times w/p)\%w, \ j) \neq 1$, **then**
        Let $f$ = random$(0, X - 1)$.
        $M_f((i + k \times w/p) \ \% \ w, \ j) = \overline{M_f((i + k \times w/p) \ \% \ w, \ j)}$.
      **If** $M_X((i + k \times w/p)\%w, \ j)) \oplus \ldots \oplus M_{n-1}((i + k \times w/p)\%w, \ j) \neq 0$, **then**
        Let $g$ = random$(X, n - 1)$.
        $M_g((i + k \times w/p) \ \% \ w, \ j)) = \overline{M_g((i + k \times w/p) \ \% \ w, \ j)}$.

4.  Repeat Steps 1–3 until all pixels are processed in $n$ meaningful shares, then return $M_0, M_1, \ldots, M_{n-1}$.

---

The time complexity of Method 4 is $O(whn)$. Figure 11 shows the flow chart of Method 4 Steps 1–3. Note that, since we made the pixels participating in the camouflage appear black when restoring the first $X$ shares and the last $Y$ shares, these two shares will look similar if $X = 2$ or $Y = 2$. See Figure 12, the experimental results of Method 4, for an example (where $N = 5$, $Y = 2$). Therefore, we give another experimental result ($N = 8$) for this method. Figure 13 shows the new experimental results.

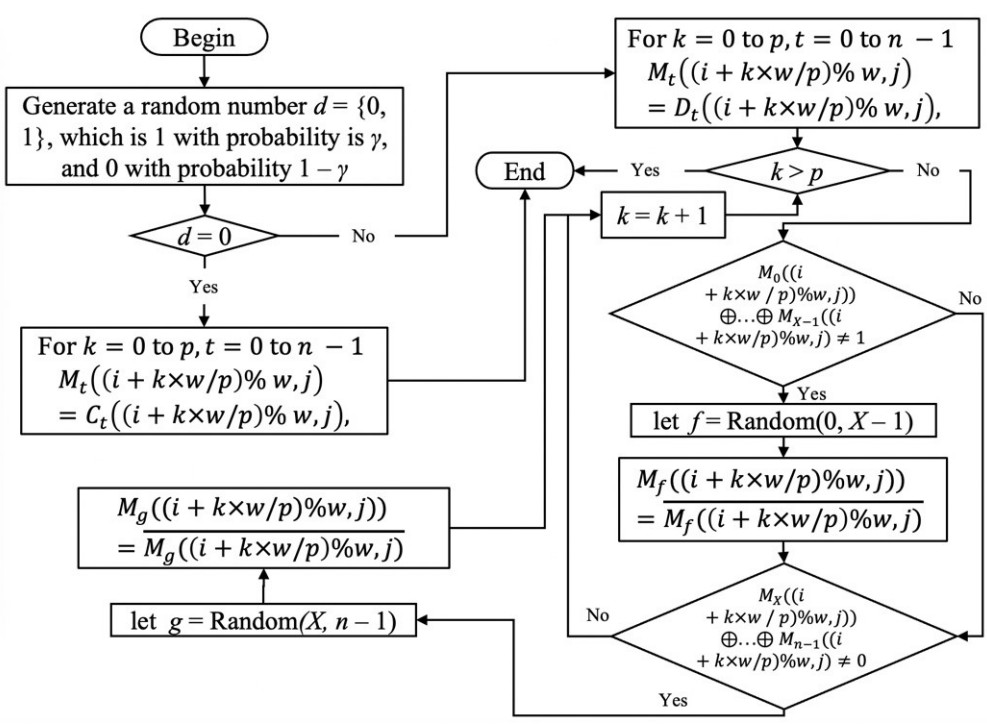

**Figure 11.** Process chart of Method 4 Steps 1–3.

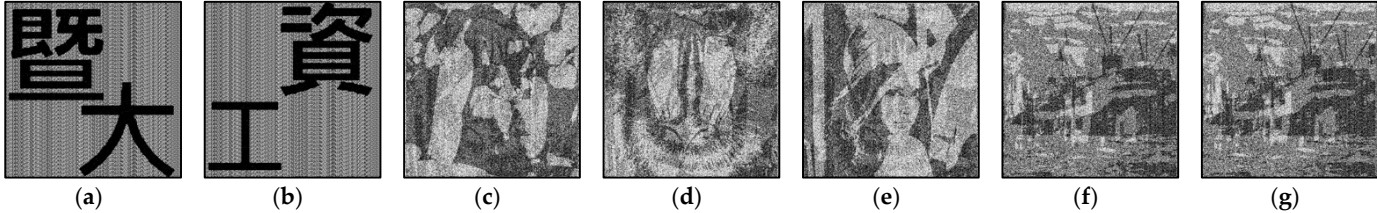

**Figure 12.** Experimental results by Method 4 for (5, 5) with $p = 45$, $\gamma = 0.5$. (**a**,**b**) Restored images $R_0$, $R_1$ (**c**–**g**) Shares $M_0, \ldots, M_4$.

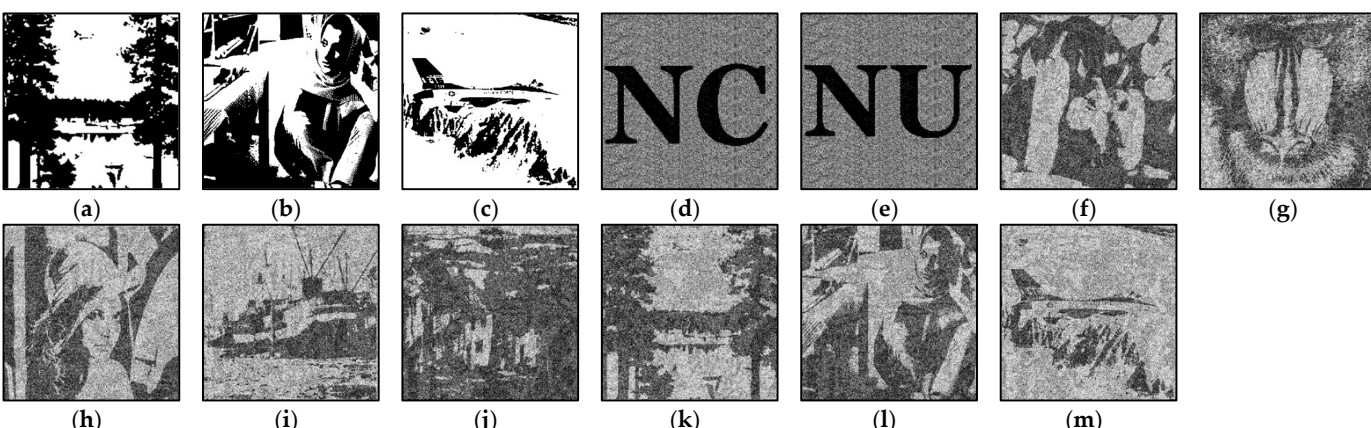

**Figure 13.** Experimental results by Method 4 for (8, 8) with $p = 10$, $\gamma = 0.5$. (**a**–**c**) Camouflage images $D_5$, $D_6$, $D_7$, (**d**,**e**) Restored images $R_0$, $R_1$ (**f**–**m**) Shares $M_0, \ldots, M_7$.

### 3.5. Algorithm IV. Secret Reconstruct Scheme

In this subsection, Algorithm IV is introduced, which decrypts the recovered image. To restore $G_1$, we pool the first $X$ shares together and apply an XOR operation to them. The remaining shares will restore $G_2$ in the same way. Similar to [8], restoring the $i$th secret

image requires shifting $G_2$'s $iw/p$ pixels and stacking them on $G_1$ by applying an XOR operation. We present Algorithm 7 for Algorithm IV as shown below. In Figure 14, the schematic is displayed.

---

**Algorithm 7.**

**Input:** $n$ meaningful shares $M_0, \ldots, M_{n-1}$ with size $w \times h$, and a positive integer $p$.
**Output:** $N$ recovered images $R_0, \ldots, R_{n-1}$ with size $w \times h$.

1.  Let $X = \lceil n/2 \rceil$.

2.  **For** each pixel in $\{(i,j) \mid 0 \leq i \leq w, 0 \leq j \leq h\}$ **do**

3.  **For** $k = 0$ **to** $N-1$ **do**
    $R_k(i,j) = M_0(i,j) \oplus \ldots \oplus M_{X-1}(i,j) \oplus M_X((i+k \times w/p)\%w, j) \oplus \ldots \oplus M_{n-1}((i+k \times w/p)\%w, j)$.

3.  **Return** $R_0, \ldots, R_{N-1}$.

---

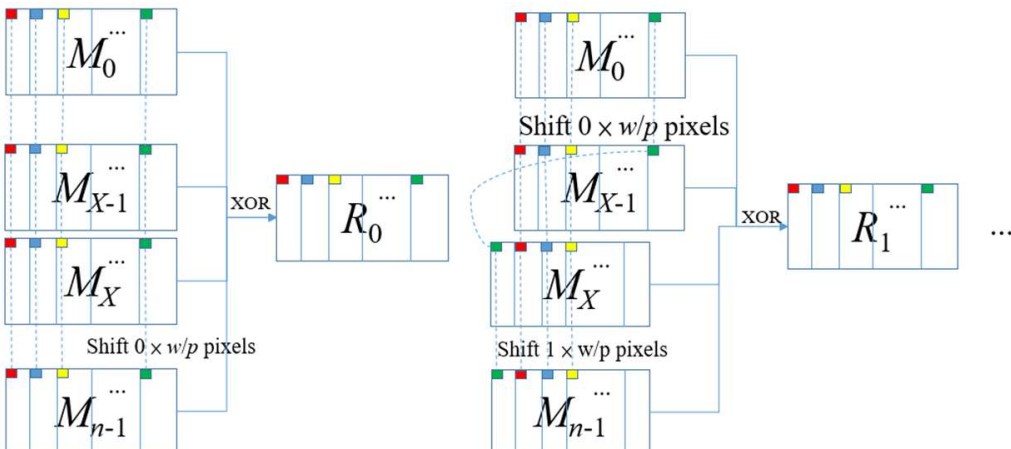

**Figure 14.** The schematic diagram of Algorithm 7.

## 4. Analysis

In this section, we perform some evaluations of the security and accuracy of the proposed schemes, and then analyze the experimental results of the proposed schemes. We discuss the contrast, PSNR, Sensitivity and SSIM, and security analysis for the proposed scheme in the following four subsections. Users can observe the changes of different methods under different values of $\gamma$.

### 4.1. Contrast Analysis

The security of a VCS and visual quality of restored images are important. We introduce the definition of *transmittance* and *contrast*, and calculate the contrast for the proposed VCS in this subsection. Usually, the value of contrast is defined as $\alpha$. The higher the contrast, the better the visual quality of the restored image. In a XOR-based VCS, the maximum possible contrast ratio is 1, which means perfect reconstruction—the restored image is exactly equal to the secret image. At first, we introduce the definition of the average transmittance. In the transmittance of an image $S$, $T(S)$ represents the proportion of white pixels in $S$, that is $T(S)$ = the number of white pixels in $S$/the total number of pixels in $S$. Allow $T[G[X(0)]]$ to stand for the average transmittance of the area in image $G$ that corresponds to the white (transparent) area in image $X$. The average transmittance of the area in image $G$ that corresponds to the black (opaque) area in image $X$ is indicated by the notation $T[G[X(1)]]$. Under this definition, we can discuss the case where $G$ = the restored image $R$, $X$ = the secret image $S$, $G$ = meaningful share $M$, and $X$ = camouflage image $D$. As

a result, the following formula can be used to determine the contrast of the restored image $R$ and meaningful share $M$.

$$\alpha_R = \frac{T[R[S(0)]] - T[R[S(1)]]}{1 + T[R[S(1)]]} \tag{1}$$

$$\alpha_M = \frac{T[M[D(0)]] - T[M[D(1)]]}{1 + T[M[D(1)]]} \tag{2}$$

We utilize each of the seven images in Figure 5 as the secret and camouflage image in turn to get more precise results. That means in total 7! = 5040 experiments were performed. Let $N = 2$, $n = 5$, and $p = 45$. Table 2 shows the average results (from 5040 experiments) for the average contrast of five shares and restored two images of the proposed Methods 1–4 when $\gamma$ is 1, 0.7, 0.5, 0.3, or 0.

**Table 2.** The Contrast analysis of the proposed schemes.

| $\gamma$ | Scheme | $\alpha_{shares}$ | $\alpha_{R_0}$ | $\alpha_{R_1}$ |
|---|---|---|---|---|
| 1 | Method 1 | 0.73112 | 0.41256 | 0.41242 |
| | Method 2 | 0.73121 | 0.00456 | 0.00370 |
| | Method 3 | 0.73105 | 1.00000 | 0.01860 |
| | Method 4 | 0.52898 | 0.00000 | 0.00000 |
| 0.7 | Method 1 | 0.46104 | 0.55675 | 0.55639 |
| | Method 2 | 0.46080 | 0.26068 | 0.25982 |
| | Method 3 | 0.46111 | 0.99480 | 0.23623 |
| | Method 4 | 0.33886 | 0.29930 | 0.29939 |
| 0.5 | Method 1 | 0.30524 | 0.66746 | 0.66728 |
| | Method 2 | 0.30492 | 0.45372 | 0.45297 |
| | Method 3 | 0.30516 | 0.99155 | 0.40721 |
| | Method 4 | 0.23194 | 0.49910 | 0.49963 |
| 0.3 | Method 1 | 0.17385 | 0.78586 | 0.78570 |
| | Method 2 | 0.17338 | 0.64508 | 0.64515 |
| | Method 3 | 0.17387 | 0.98794 | 0.60569 |
| | Method 4 | 0.12970 | 0.69844 | 0.69816 |
| 0 | Method 1 | 0.00022 | 0.98313 | 0.98359 |
| | Method 2 | 0.00011 | 0.98335 | 0.98342 |
| | Method 3 | 0.00035 | 0.98344 | 0.98328 |
| | Method 4 | 0.00012 | 0.98347 | 0.98350 |

As $\gamma$ decreases, the contrast value of the shares decreases while the contrast of the restored images increases, as seen in Table 2. $\alpha_{shares}$ in Methods 1–3 is almost the same and better than $\alpha_{shares}$ in Method 4. The contrast of the two restored images is about the same when the value of $\gamma$ is the same for Methods 1, 2, and 4. In the case of Method 3 with the same value of $\gamma$, the contrast of the first restored image (almost 1) is much better than that of the second restored image. In conclusion, the average contrast of the restored images of Method 1 gives the best results. Method 3 has the best contrast of the first restored image, while the contrast of the second restored image is worse than that of the other methods.

### 4.2. PSNR Analysis

In this subsection, we introduce the Peak Signal-to-Noise Ratio (PSNR), which is a ratio of the maximum possible power of the signal and destructive noise power affecting its representation accuracy, and PSNR is often expressed in decibel units. In the image, we can use PSNR as a more objective method to calculate the distortion of the image because it has quantitative data. The PSNR is defined simply by the mean-square error (MSE). Given an original image $I$ of size $w \times h$ and a reconstructed image $K$, the PSNR and mean square error (MSE) are defined as:

$$MSE = \frac{1}{w \times h} \sum_{0}^{m-1} \sum_{0}^{n-1} [I(i,j) - K(i,j)]^2 \tag{3}$$

$$PSNR = 10 \times \log_{10}\left(\frac{MAX_I^2}{MSE}\right) \tag{4}$$

$MAX_I$ is the maximum possible value of pixels for an image. If each pixel is represented by 8 bits, then black pixels are 0 and white pixels are 255, so $MAX_I$ in our scheme is always calculated using 255. We perform PSNR analysis of our schemes in Table 3. In this table, we use the corresponding method to encrypt two secret images and five camouflaged images with $p = 45$, $N = 2$, and $n = 5$ in different $\gamma$ cases for the analysis. The values in this table are the average PSNR for two restored images (except Method 3) and five shares. A total of 7! (= 5040) experimental results were averaged and presented in Table 3. PSNR is generally used for gray-scale image or color image analysis, so the results are for reference only since we use binary data in these schemes.

**Table 3.** The PSNR analysis of the proposed schemes.

| $\gamma$ | Image | Method 1 | Method 2 | Method 3 * | | Method 4 |
|---|---|---|---|---|---|---|
| 1 | shares | 10.08540 | 10.08555 | 10.08565 | | 7.21653 |
| | Restored image | 6.12667 | 3.01978 | inf | 3.03151 | 1.10599 |
| 0.7 | shares | 6.59821 | 6.59845 | 6.59842 | | 4.64744 |
| | Restored image | 7.60584 | 4.59838 | 27.66252 | 4.66005 | 2.65272 |
| 0.5 | shares | 5.23130 | 5.23122 | 5.23126 | | 4.61973 |
| | Restored image | 9.03888 | 6.07030 | 25.54497 | 6.08073 | 4.10810 |
| 0.3 | shares | 4.22439 | 4.22448 | 4.22460 | | 3.88114 |
| | Restored image | 11.15357 | 8.10970 | 23.99535 | 8.19519 | 6.36929 |
| 0 | shares | 3.01179 | 3.01144 | 3.01125 | | 3.01124 |
| | Restored image | 22.54548 | 22.54521 | 22.55623 | 22.55617 | 22.5452 |

* The PSNR values of the restored images are different in Method 3. The restored image on the left in this column is the first image, and on the right is the second image.

Observing Table 3, we can find that the PSNR value of the shares decreases as $\gamma$ decreases. The PSNR values of shares in Methods 1–3 are almost the same and better than shares in Method 4. As $\gamma$ decreases, the PSNR of restored images gets better. With the same value of $\gamma$ for Method 1, Method 2, and Method 4, the PSNR of the two restored images is almost identical, so we show their average value in this table. In Method 3 with the same value of $\gamma$, the PSNR for the first restored image is much better than the second restored image; thus, we write them separately. In conclusion, Method 1 gives the best results for the average PSNR in restored images, and Method 3 has the best PSNR for the first restored image.

*4.3. Sensitivity and SSIM Analysis*

PSNR is mainly for calculating the image distortion rating data, and sometimes it does not fully represent the human visual perception. Numerous experimental findings indicate that the visual quality perceived by the human eye and PSNR numbers do not always match up completely. In fact, higher PSNR scores may appear worse than lower PSNR scores. Therefore, we will introduce Sensitivity in this subsection. In the sensitivity analysis, we focus on the restoration of black pixels, called a black pixel in a restored image positive; $N_{TP}$ and $N_{FN}$ represent the number of pixels with the "true positive" and "false negative" features in the restored image. A pixel that is black in the original image is called a "true positive" pixel if it is still black after restoration; on the other hand, a pixel that is black in the original image is called a "false negative" pixel if it is restored to white. The formula for Sensitivity is given below:

$$Sensitivity = \frac{N_{TP}}{N_{TP} + N_{FN}} \tag{5}$$

We perform Sensitivity analysis for each of our proposed methods with $p = 45$, $N = 2$, and $n = 5$ in different $\gamma$, then present the results together. A total of 7! (= 5040) experimental results were averaged and given in Table 4.

**Table 4.** The Sensitivity analysis of the proposed schemes.

| $\gamma$ | Image * | Method 1 | Method 2 | Method 3 | Method 4 |
|---|---|---|---|---|---|
| 1 | $R_0$ | 0.75828 | 0.80167 | 1.00000 | 1.00000 |
| | $R_1$ | 0.75806 | 0.80166 | 0.51906 | 1.00000 |
| 0.7 | $R_0$ | 0.82518 | 0.85985 | 0.99821 | 0.99846 |
| | $R_1$ | 0.82533 | 0.85961 | 0.66205 | 0.99829 |
| 0.5 | $R_0$ | 0.87733 | 0.89863 | 0.99709 | 0.99723 |
| | $R_1$ | 0.87721 | 0.89842 | 0.75523 | 0.99726 |
| 0.3 | $R_0$ | 0.92243 | 0.93629 | 0.99585 | 0.99591 |
| | $R_1$ | 0.92264 | 0.93640 | 0.84933 | 0.99598 |
| 0 | $R_0$ | 0.99438 | 0.99441 | 0.99445 | 0.99449 |
| | $R_1$ | 0.99450 | 0.99442 | 0.99443 | 0.99456 |

* $R_0$ and $R_1$ represent the first and the second restored image, respectively.

The structural similarity index (SSIM) is a technique for calculating how two similar images are in terms of structure, contrast, and luminance. SSIM takes the value range [0, 1], and the higher the value, the lower the image distortion. Compared with the traditional image quality measurement standards, SSIM is more consistent with the human vision judgment of image quality. Given an original image $I$ of size $w \times h$ and a reconstructed image $K$, we calculate the average value $E(I)$ in image $I$, the average value $E(K)$ in image $K$, and the three variances $S_I$, $S_K$, $S_{IK}$ by using the formula demonstrated below. A constant term $c_1$ is set equal to $(K_1 \times L)^2$ and $c_2$ is set equal to $(K_2 \times L)^2$. Set the black pixel to 0, white pixel to 255, $K_1$ to 0.01, $K_2$ to 0.03, and $L$ to 255, to avoid a denominator of 0 and keep it stable. The formula of the SSIM is given below:

$$SSIM(I, K) = \frac{2 \times E(I) \times E(K) + c_1}{E(I)^2 + E(K)^2 + c_1} \times \frac{2 \times S_{IK} + c_2}{S_I^2 + S_K^2 + c_2} \tag{6}$$

$$E(I) = \frac{1}{w \times h} \sum_{i=1}^{w \times h} I_i \tag{7}$$

$$E(K) = \frac{1}{w \times h} \sum_{i=1}^{w \times h} K_i \tag{8}$$

$$S_I = \sqrt{\frac{1}{w \times h} \sum_{i=1}^{w \times h} (I_i - E(I))^2} \tag{9}$$

$$S_K = \sqrt{\frac{1}{w \times h} \sum_{i=1}^{w \times h} (K_i - E(K))^2} \tag{10}$$

$$S_{IK} = \frac{1}{w \times h} \sum_{i=1}^{w \times h} (I_i - E(I)) \times (K_i - E(K)) \tag{11}$$

We perform SSIM analysis for each of our proposed methods with $p = 45$, $N = 2$, and $n = 5$ in different $\gamma$, then present the results together. A total of 7! (= 5040) experimental results were averaged and given in Table 5.

**Table 5.** The SSIM analysis of our schemes.

| $\gamma$ | Image * | Method 1 | Method 2 | Method 3 | Method 4 |
|---|---|---|---|---|---|
| 1 | $R_0$ | 0.51361 | 0.00633 | 1.00000 | 0.00000 |
| | $R_1$ | 0.51371 | 0.00623 | 0.02929 | 0.00000 |
| 0.7 | $R_0$ | 0.65416 | 0.29760 | 0.99657 | 0.22111 |
| | $R_1$ | 0.65418 | 0.29742 | 0.30329 | 0.21992 |
| 0.5 | $R_0$ | 0.75793 | 0.49789 | 0.99442 | 0.45613 |
| | $R_1$ | 0.75782 | 0.49791 | 0.50376 | 0.45631 |
| 0.3 | $R_0$ | 0.84490 | 0.69740 | 0.84933 | 0.68785 |
| | $R_1$ | 0.84469 | 0.69795 | 0.69750 | 0.68774 |
| 0 | $R_0$ | 0.98884 | 0.98885 | 0.98889 | 0.98889 |
| | $R_1$ | 0.98902 | 0.98892 | 0.98887 | 0.98897 |

* $R_0$ and $R_1$ represent the first and the second restored image, respectively.

Observing Tables 4 and 5, we can find that in Methods 1, 2 and 4, as $\gamma$ decreases, the sensitivity and SSIM value of the restored images becomes better and are almost identical. In turns of sensitivity, Method 4 gives the best result; Method 2 is slightly better than Method 1. In Method 3, as $\gamma$ decreases, the sensitivity of the first restored image becomes slightly worse, but the second restored image becomes better, while the first recovered image has a sensitivity close to 1 and is always better than the second recovered image. In conclusion, the average sensitivity result of the restored image for Method 4 is the best. Method 1 has the best average SSIM results for the restored images. Method 3 has the best SSIM for the single restored image.

*4.4. Security Analysis*

In the proposed scheme, we need to stack all shares to fully recover the original secret image; otherwise, the secret image cannot be leaked. For each pixel, it is only encrypted for two consecutive secret images, or is used as a camouflage image. When the pixel is used as a camouflage image, if we stack only $k$ ($k < n$) shares, it just like we XOR $k$ random values by our Algorithm III, so the original secret image cannot be reconstructed. It also produces unpredictable images because it is impossible to determine whether the stacked pixels are black or white. Therefore, it is impossible to correctly recover any secret image without collecting all $N$ shares when $\gamma \neq 0$. On the other hand, if we collect all of the first $n/2$ shares (or the last $n/2$ shares) and stack them using the XOR operation, $G_1$ (or $G_2$) will be recovered if $\gamma = 0$. Note that, from the constructing of Algorithm 1, we seem to be

sacrificing some security in the pursuit of optimizing the contrast of the restored image. From Algorithm 1, it can be found that if we XOR the output $G_1$ or $G_2$ with a version of itself shifted over by $w/p$ pixels, then we get an image where most of the pixels depend only on the secret images: $G_1(i, j) \oplus G_1(i + w/p, j) = S_A(i + w/p, j) \oplus S_{A+1}(i, j)$ and $G_2(i + Aw/p, j) \oplus G_2(i + (A+1)w/p, j) = S_A(i, j) \oplus S_{A+1}(i, j)$. The latter one implies $G_2(i, j) \oplus G_2(i + w/p, j) = S_A(i–Aw/p, j) \oplus S_{A+1}(i–Aw/p, j)$. However, since $A$ is randomly selected from 0 to $N$–1, it is impossible to tell which pixel is encrypted by which two consecutive secret pixels. This means that the stacked image will be a complex image where all the secret images and their shifted images are interleaved together; there are $2N$ possibilities, so it is still impossible to identify any single secret image. Even taking into account the edge case where all but one of the secret images ($A$) are completely 0, this still does not fully reveal the remaining secret image—there are still two possibilities: $G_1(i, j) \oplus G_1(i + w/p, j) = S_A(i + w/p, j)$ or $S_A(i, j)$ and $G_2(i, j) \oplus G_2(i + w/p, j) = S_A(i – Aw/p, j)$ or $S_A(i – (A–1)w/p, j)$. That means it will appear as a mixed image of the secret image $A$ and itself shift by $w/p$ pixels. However, this is still an unwelcome result. A good choice when there are very few secret images is to use Method 1 and set $\gamma = 1$. In this case, an influence of the Algorithm 1 will disappear, and the problems that may be caused can be solved at the same time. The original secrets can still be restored, but the contrast of the restored image cannot be as high as when using Algorithm 1 ($\gamma < 1$) (see Table 2). Fortunately, Algorithms II and III (Algorithms 3, 4, 5 or 6) can generate interference and increase the uncertainty of shares and make leaks less likely, provided that $\gamma$ cannot be equal to 0. Since the proposed scheme uses Algorithms I, II and III (Algorithms 3, 4, 5 or 6) completely sequentially, the user cannot obtain $G_1$ or $G_2$ directly.

Under such observation, the value of $p$ cannot be set too small, the value of $\gamma$ cannot be too close to 0, and avoid setting all but one secret image to white (in the same position). In addition, KK2 and KK3 can be selected for $f_{XRG}$ and $f_{XORG}$ instead of KK1 for Algorithm I. This will avoid the above problem, but the performance of the new algorithm will not be as good as it currently is.

## 5. Concluding Remarks

This paper studies the ($n$, $n$) XOR-based multi-secret sharing schemes with meaningful sharing. There are four different methods held in Algorithm III. The shares and restored images produced by these four methods are different. It can be said that we have designed four different VCSs. At the same time, there are two parameters $\gamma$ and $p$ in the proposed VCSs that will affect the light transmittance of the shares and restored images. These two values can also be determined by the user, which makes the proposed VCSs more flexible. Therefore, the user can decide the appropriate method and parameters according to the situation.

The time complexity of the encryption process of the proposed VCSs is O($whn$), and the time complexity of the decryption process is O($whnN$). Theoretical analysis and experimental results show that the proposed VCSs are safe and effective. All our proposed VCSs have the following characteristics:

1. No pixel expansion.
2. Multiple secrets can be encrypted at the same time.
3. Each share can be disguised as a different meaningful image.
4. Both shares and reconstructed images have good visual quality.
5. Parameters $\gamma$ and $p$ can be adjusted as required.

Table 6 shows the comparison of the proposed scheme with some previous research. Among these four proposed methods, according to the analysis in Section 4, if the user needs to obtain one of the clearer restored images, then Method 3 is the best recommendation; otherwise, Method 1 performs better than other methods in all aspects. However, Method 4 can obtain perfect black restoration, which is helpful for visual recognition of the restored image, and is also a recommended method when $N$ and $n$ are large.

**Table 6.** Compare the proposed scheme with related works.

| Schemes | Features | | | |
|---|---|---|---|---|
| | **Without Pixel Expansion** | **Meaningful** | **Multi-Secret** | **XOR-Based** |
| [1] (1994) | No | No | No | No |
| [2] (1987) | Yes | No | No | No |
| [6] (2013) | Yes | Yes | No | Yes |
| [7] (2015) | Yes | Yes | No | Yes |
| [8] (2021) | Yes | Yes | No | Yes |
| [9] (2018) | Yes | No | Yes | No |
| [11] (2008) | Yes | Yes | No | No |
| [13] (2020) | Yes | Yes | Yes | No |
| [14] (2019) | Yes | Yes | No | No |
| Ours | Yes | Yes | Yes | Yes |

The VCS presented in this paper has many topics for further study. For example, how to enhance the proposed VCSs to apply to grayscale or color images, modify the proposed VCSs to extend the encrypt ability of $(n, n)$-threshold to $(k, n)$-threshold for any $k < n$, and consider whether the ability of the proposed VCSs can be increased, so that the OR operation can also be used to recover the secret image. Besides, since we initially encrypt the $N$ secret images into two shares in Algorithm I, although these two shares are subsequently encrypted into $n$ shares in Algorithm II, this process adds distortion of the schemes. Therefore, it will be fascinating to work on in the future to directly encrypt $N$ secret images into $n$ shares in order to reduce distortion.

**Author Contributions:** Conceptualization, J.S.-T.J.; methodology, J.S.-T.J. and S.-Y.H.; software, S.-Y.H.; validation, S.-Y.H., A.-h.L. and J.S.-T.J.; formal analysis, S.-Y.H. and J.S.-T.J.; investigation, J.S.-T.J.; resources, S.-Y.H.; data curation, S.-Y.H.; writing—original draft preparation, S.-Y.H. and A.-h.L.; writing—review and editing, J.S.-T.J.; visualization, S.-Y.H. and A.-h.L.; supervision, J.S.-T.J.; project administration, J.S.-T.J.; funding acquisition, J.S.-T.J. All authors have read and agreed to the published version of the manuscript.

**Funding:** This research was funded by the Ministry of Science and Technology of Taiwan, ROC, Grants number MOST 110-2221-E-260-003, and 111-2115-M-260-001.

**Institutional Review Board Statement:** Not applicable.

**Informed Consent Statement:** Not applicable.

**Data Availability Statement:** Not applicable.

**Acknowledgments:** The authors would like to thank the reviewer for the constructive feedback.

**Conflicts of Interest:** The authors declare no conflict of interest.

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
