# Peer review of "XOR-Based Meaningful (n, n) Visual Multi-Secrets Sharing Schemes"

_applsci, doi:10.3390/app122010368_

Round 1

Reviewer 1 Report

Will be fascinating to work on in the future to directly encrypt 528 N secret images into n shares in order to reduce distortion; and enhance the proposed VCSs to apply to grayscale or color images.

Author Response

We would like to thank the referees for their careful reading of the manuscript and fruitful comments. We already modify and rewrite our manuscript carefully.

Reviewer 2 Report

what i like in this article is every pseudo code explain with flowchart and sample image, this is more important so reader will understand what author meaning. keep it work.

Author Response

(The authors gave the same response as above.)

Reviewer 3 Report

This paper creates multiple shares that can be decrypted into one of several images by offsetting the shares and overlaying them.  There are three main algorithms.  Algorithm 1 comes from [8] mostly unchanged, which generates 2 shares G1 and G2 from N images.  Algorithm 2 is a standard secret sharing scheme applied twice to encode G1 and G2 into n shares (each is encoded into n/2 shares).  Algorithm 3 is the new bit, which modifies the shares by randomly mixing in camouflage images so that each share looks like a noisy version of the original camouflage image.

I'm a bit worried about the idea of using XOR.  The entire point of visual cryptography is that decryption can happen without any machine assistance and very easily.  However "XOR-based RGVCS needs some lightweight, small, and cheap computing devices."  This seems to defeat the purpose of visual cryptography.  Once you allow enough computing power to do XOR on an image (getting an image into and out of a device is already pretty non-trivial!) there are many many things that you can do with traditional cyrptography, including steganography and secret sharing, which appear to be the main purposes of this paper.  In short, it's not clear why this work is beneficial to anyone.

Critically, security is not addressed in this paper, and it is relatively straightforward to show that the scheme is insecure, as is the scheme in [8].  We will assume that the image indices are all taken modulo w and module h respectively, set v = w / p, and take pixel addition modulo 2 (these would have been a great benefit to readability in the paper!)  Now look at step 4 in Algorithm 1.  Consider two consecutive values of k.  We have:
G2[ i + (A + k + 1)v, j] = S_{A+1}[i + kv, j] + G1[i + kv, j]
and
G2[ i + (A + k + 2)v, j] = S_{A+1}[i + (k+1)v, j] + G1[i + (k+1)v, j]
substitute in for g1 to get
   = S_{A+1}[i + (k+1)v, j] + S_A[i + (k+1)v, j] + G2[ i + (A + k + 1)v, j]

Now add these two pixels together and we get:
S_{A+1}[i + (k+1)v, j] + S_A[i + (k+1)v, j]

and so if we XOR G2 with a version of itself shifted over by v pixels then we get an image where most of the pixels (it isn't clear what is going on it step 5) depend only on the images, and have no randomness involved.  Considering an edge case where all but one of the images are entirely 0 then doing this will reveal (a random subset of) the pixels in the other image.  Something similar also happens with G1.

So algorithm 1 is completely insecure.

Algorithm 2 uses standard techniques to encode G1 and G2 separately into n/2 shares each.  As noted above, either the G1 half of the shares or the G2 half of the shares will leak information.  So this is at best a n/2 out of n threshold scheme.

It appears that Algorithm 3 is intended to disguise the shares as innocuous images.  This process is called steganography.  There is a large body of literature on steganography which is not cited in this work.  The techniques used in this paper would be familiar to steganographers.

Overall the paper is poorly written.  The algorithms have very little explanation and are presented in an obscure way (why do we need f_xorg?  It is just XORing two bits.)  Much of this seems to have been preserved from [8] which also has very little explanation.

Author Response

We would like to thank the referees for their careful reading of the manuscript and fruitful comments. We already modify and rewrite our manuscript carefully. The responses to each comment can be found right the comment in the attachment.

Round 2

Reviewer 3 Report

I haven't seen much improvement in my main concerns here.  Replying to the reply:

1. The additional information somewhat helps out.  I'm still not sure why this specialized polarization device is any better than, say, a smart phone.  I'll leave this for now since this is not the most concerning part of this paper.

2. The standard for papers proposing new cryptographic protocols/primitives is to have a security proof (eg for something like the one time pad) or other arguments explaining how the protocol/primitive defeats known attacks (eg something like AES).  There is barely any of that here, just the paragraph attempting to address one my criticisms.  About that, the fact that we can't tell which image a pixel came from is irrelevant.  Information is leaked quite readily with this system.  For a demonstration in a similar context, have a look at https://www.douglas.stebila.ca/teaching/visual-one-time-pad/ and click the "move to ciphertext 1" button below ciphertext 2.  You can very clearly see the two plaintext images superimposed on one another.  This also has the property that we can't tell which pixel in the resulting image came from which original image, but it doesn't matter.  An enormous amount of information is still leaked.  In this article you will get one image superimposed on a shifted copy of itself.  That still leaks plenty of information. 

The later stages do in fact complicate things, but the information has already been leaked and the later stages can still allow this through.  As already established, the first n/2 shares from algorithm 2 are sufficient to reconstruct G1 and hence obtain a large amount of information in the images.  All of the methods for algorithm 3 include cases (d = 0) when the shares from algorithm 2 are copied directly.  So we just need the first n/2 final shares and perform the same attack; any pixel created by a d=0 case in algorithm 3 will leak information.  In method 1 when d=1 se see that the first n/2 shares are doctored so that they XOR together to get S_Z, so in that case again we can obtain information from only n/2 shares.  Remember that this is supposed to be an n out of n secret sharing scheme!

3. The method of hiding the shares in camouflage images is absolutely steganography.  There is much more to it that just encoding a message in a single image.  It refers to any techniques where the intent is to conceal the existence of a message (here, the shares).  Is that not the intent here?  If not, then what are the camouflage images camouflaging?  You should cite some appropriate references in the steganography literature.

4. Some explanatory text was added, but the algorithms are still needlessly complicated and not very well explained.

Overall I believe that the authors are not very aware of what is required in cryptography research.  There needs to be a substantial amount of effort put towards the security of the scheme, indeed the security of the scheme is always the priority.  That is not the case in this paper.

My suggestion: start again and instead of going to 2 shares, go straight to n shares to avoid the n/2 attacks.  In step 4 of algorithm 1, figure out something different.  You need to get more randomness in there.  This is basically some variant of OTP and you need to have at least as much key material as the plaintext to get security, and if you want to have n shares then you need (n-1) times as much randomness as the original plaintext (here, number of pixels).  Whether you can make it work and still get the cyclic shift thing I don't know.  After getting the shares you can do what you want to hide them, but don't refer back to the original images (eg case d=1 in algorithm 3), just encode the shares.  I suggest looking into the steganographic literature for that one.  Keep in mind that the security for steganography often assumes that the attacker has access to (an approximate copy of) the camouflage images!  A couple of the Algorithm 3 methods would leak plaintext information in that case.

Author Response

We thank the reviews for their time spent on our manuscript. Sorry, our previous revision mistakenly turned off the "Track Changes” function, so the review couldn’t quickly see where we made changes. This time we are more careful with this part. We hope that your concerns can be addressed in this mail. The detailed reply is as the attached file.
